# Comparing descriptive and theoretical models of decision-making under uncertainty and their relation to socioeconomic factors

Brendan Lam[1]*, Samuel Paskewitz[1], Hunter Robbins[1], Arielle Baskin-Sommers[1,2]

1 Psychology Department, Yale University, New Haven, Connecticut, United States of America,
2 Psychiatry Department, Yale University, New Haven, Connecticut, United States of America

* brendan.lam@yale.edu

## Abstract

This study examines the selection and validation of measurement models for decision-making under uncertainty, with particular emphasis on the integration of socioeconomic contexts in these models. We critically compared four distinct models, differing in their mathematical form of risk and ambiguity, to determine which best predicts willingness-to-pay behavior in a financial decision-making task. We used Bayesian hierarchical modeling to fit each measurement model and the Leave-One-Out Information Criteria to assess how each model performed. In a sample of 74 community members, we found that the maximal descriptive model — one that accounts for the variance across participants' sensitivity to changes under risk and ambiguity — demonstrated superior predictive accuracy in out-of-sample testing. Notably, our results revealed that adding socioeconomic factors into the model improved prediction. Further, people from higher-income households exhibited a greater aversion to ambiguity, whereas those from lower-income households showed less aversion to ambiguity. There was a lack of association between annual household income and risk. These findings not only highlight the importance of rigorously validating measurement models but also underscore the pivotal role of socioeconomic factors in shaping decision-making under uncertainty. Future research should further investigate how aspects of socioeconomic background—such as childhood economic conditions and environmental unpredictability—influence decision-making. It also is important to test whether ambiguity aversion estimates, when contextualized by socioeconomic variables, can reliably predict real-world decision-making under uncertainty.

## Introduction

Uncertainty in our everyday lives is omnipresent. Just in the financial domain alone there are potential daily occurrences of having to make decisions with incomplete

**Data availability statement:** All data files are available from our Open Science Framework repository: https://osf.io/4hfcs/.

**Funding:** This research was supported in part by grants through the American Psychological Foundation, American Psychology-Law Society, and Harry F. Guggenheim Foundation (PI: Baskin-Sommers).

**Competing interests:** The authors have declared that no competing interests exist.

or imperfect knowledge about an event, outcome, or situation. People may decide whether to make a particular investment, with the outcome potentially being a gain or a loss depending on the market. People may decide whether to make a large purchase, while considering if they will use the item enough to justify the expense or if they should save the money for future needs or emergencies instead. People may decide about purchasing groceries on a given week, not knowing if they will have enough money to cover urgent costs like medical bills or whether public assistance will be received in time to cover their needs. When making decisions such as those above, people must decide the best course of action based on the limited information they have at the moment, navigating the inherent uncertainty of the decision. Researchers distinguish two types of uncertainty: risk and ambiguity [1,2]. In decisions under risk, the probability of each outcome is known (e.g., knowing there is a 1% chance of winning the lottery from a single lottery ticket). In contrast, decisions under ambiguity entail choosing when the outcome probabilities are unknown and undiscoverable through logical deduction or inductive inference (e.g., drawing a red ball out of a jar when one does not know how many balls are in the jar) [3]. Mathematical measurement models of uncertainty distinguish between parameters for risk propensity and ambiguity aversion [4–7], and these models can be expressed in various forms, each using different functions and parameters to describe the processes of risk propensity and ambiguity aversion [8,9]. The plurality of these measurement models presents opportunities to test and compare different conceptualizations of decision-making under uncertainty.

Two types of measurement models currently dominate the psychological literature: a linear and an exponential subjective value model [5,10,11]. Both models are based on subjective expected utility theory, which posits that a decision maker assigns a utility to each option and weighs the utility by the probability of the event occurring [12]. The linear and exponential subjective value models have one component that represents the utility calculation and another component that represents the perceived probability of each event. Thus, both are theoretical models that formalize subjective expected utility theory. In the linear subjective value model [11], a decision-maker adjusts the perceived probability of an outcome based on 1) the amount of risk present, 2) the amount of ambiguity present, and 3) their aversion or approach toward ambiguity. Similar to the linear subjective value model, the exponential subjective value model calculates the expected utility by estimating a decision-maker's ambiguity aversion and risk propensity [5]. Risk propensity is modeled in the same way in the exponential subjective value model as the linear subjective value model. However, unlike the linear subjective value model, the exponential subjective value model estimates the perceived probability of winning as an exponential function of ambiguity aversion and the ambiguity amount [5]. In both models, people on average demonstrate themselves to be risk and ambiguity averse; that is, they are more conservative in decision-making under riskier and more ambiguous conditions [13,14].

Previous research has used theoretical models to investigate the neural and behavioral correlates of risk and ambiguity [2,15,16]. However, more broadly within the cognitive modeling literature, there are strategies that are more descriptive, such

as summarizing patterns in the data (e.g., means, covariances) without positing any theoretical mechanisms behind these patterns [17]. This descriptive approach to modeling risk and ambiguity has been used less frequently in the literature. Generally, a descriptive model can be viewed as a special case of the theoretical models, as descriptive models use the same variables and task data [18–20]. However, they differ in their functional form, such as whether probability varies as a linear function of ambiguity [7,21]. Descriptive models include linear regression, as such, these models can show the influence of one variable (e.g., ambiguity) over and above other variables (e.g., risk). More specifically, the maximal model contains all effects from an experiment (i.e., main effects and interactions) as well as all within-subject and within-item variance components. Of note, theoretical models do not have an intercept or interaction term between risk and ambiguity, meaning they do not account for the simultaneous presence of risk and ambiguity in a person's decision. Thus, a descriptive model can show how a person's decision varies as a linear function of risk, ambiguity, and the combination of risk and ambiguity. Including the maximal descriptive model in the model-building process is recommended to provide an interpretable baseline model to compare to other theoretically derived models [19]. Without a baseline model that is "theory-free," researchers run the risk of creating complex models that may be more difficult to fit and interpret, and that make assumptions about decision-making without thoroughly testing alternatives. Thus, one goal of the present study was to test and compare descriptive (e.g., maximal model) and theoretical models (e.g., linear subjective value model) for estimating decision-making under uncertainty.

Testing alternative measurement models in decision-making under uncertainty provides an opportunity for refining the estimation of key decision processes. Beyond mathematical estimation, the influence of external conditions on people's decisions is an important factor to consider. One condition that has been studied widely in decision-making research broadly is socioeconomic factors [22–24] (see [23] for review). However, there has been relatively less research on the influence of socioeconomic factors on the decision-making processes of risk or ambiguity. One study on risk specifically, finds that lower subjective socioeconomic status, as measured by peoples' self-reported placement on a socioeconomic ladder, is associated with greater risk aversion, as measured by a hypothetical monetary gambling task. This suggests that people with fewer resources are more hesitant to risk losing what they already possess [25]. Another study of risk finds that childhood deprivation, as measured by perceived social and economic status, predicts greater risk aversion in the balloon analog risk task [26]. The authors contend that early life socioeconomic status influences people's preference for minimizing uncertainty (see also [27]).

Research on the relationship between socioeconomic factors and ambiguity aversion has been mixed [28,29]. For example, studies using hypothetical monetary gambles or hypothetical financial investments in samples of Dutch participants (with average annual household incomes of €45,792 and €35,256, respectively) [30,31] show that individuals with higher economic resources display lower ambiguity aversion [30,31] (i.e., decreasing absolute ambiguity aversion), that is, people with more wealth appear more willing to make decisions with uncertain outcomes (e.g., investing in education or the stock market) for potentially higher returns [32]. However, research conducted in developing countries using ambiguous phrases, field experiments, or a two-choice decision task tends to find different patterns for the relationship between ambiguity and decision-making depending on the experimental context [28,29]. One study using ambiguous phrases shows that people living in urban areas with lower economic resources were more ambiguity seeking, whereas those living in rural areas with lower economic resources were more averse to ambiguity [29]. Importantly, though, the effects were modified by the relative level of household income, such that within lower economic resourced families, those that had more resources show more sensitivity to ambiguity. This suggests that for the poorest people, the constant exposure uncertainty [33] may lead them to be less adverse to ambiguous probabilities [34]. Other studies using field experiments (e.g., distribution of crops) or a two-choice decision task show that relative income level among those in developing countries did not relate to ambiguity sensitivity [28]. To date, though, no studies first compared different measurement models of decision-making before testing the relationship between socioeconomic resources and decision-making under uncertainty. This means that the mixed findings within this literature could, in part, reflect a failure to accurately estimate the decision processes of risk and ambiguity when put into the context of real-world resources.

The purpose of the present study is to build, compare, and assess different measurement models of decision-making under uncertainty (i.e., risk and ambiguity) and understand their association with socioeconomic factors. We use a financial decision-making task in which people make decisions under varying amounts of risk and ambiguity. First, we explore which measurement model best accounts for people's decisions. We test both descriptive models of decision-making under uncertainty and theoretical models. Using a hierarchical Bayesian analysis framework, we compare and evaluate all models with the leave-one-out information criterion. Second, we investigate the relationship among different dimensions (i.e., annual household income and neighborhood disadvantage) of socioeconomic factors with risk propensity and ambiguity aversion. Given the relevance of both household and community-level [35] socioeconomic factors on decision-making, we investigate the influence of annual household income and neighborhood-level disadvantage on risk and ambiguity. Consistent with previous literature, we hypothesize that lower household income and greater neighborhood disadvantage would be associated with greater aversion to risk. Given that previous research on ambiguity aversion shows a mixed pattern of results [28–31,36], we did not have specific hypotheses for this process. Findings from the present study have the potential to deepen our understanding of the utility of various decision-making models under uncertainty and clarify how socioeconomic factors may shape risk and ambiguity processes.

## Materials and methods

### Participants

59 male (79.7%) and 15 female (20.3%) adults aged 18–55 (M = 36.59, SD = 11.74) were recruited from the community between 12/6/2014 and 7/10/2015 through flyers in New Haven County, Connecticut. Our recruitment in a wide array of areas in New Haven County resulted in a sample that contained a range of people with different socioeconomic backgrounds. Almost half of the participants in the sample (46.0%) were unemployed, while the remainder were employed either full-time or part-time (36.5%), full-time students (12.2%), or on disability (5.3%). Educational attainment was as follows: 27% high school diploma, GED, or less; 59.5% vocational school, some college or bachelor's degree; and 13.5% graduate work or degree. Most participants self-identified as Black/African American (52.7%) or as White (44.6%), with the remainder of the sample identifying as mixed racial identity (1.4%) or Asian (1.4%). See the Results section for more details about the socioeconomic variables.

A prescreen phone interview and in-person assessment materials were used to exclude people who were younger than 18 or over 55, had performed below the fourth-grade level on a standardized measure of reading (Wide Range Achievement Test-III) [37], who scored below 70 on a brief measure of IQ [38], who had diagnoses of schizophrenia, bipolar disorder, or psychosis, not otherwise specified [39], or who had a history of medical problems (e.g., uncorrectable auditory or visual deficits; head injury with loss of consciousness greater than 30 minutes) that may impact their comprehension of the materials or performance on the task. Participants were excluded if they showed no behavioral variability in the decision-making under uncertainty task. All participants provided written informed consent. All procedures were completed in line with the protocol approved by the Yale University Institutional Review Board (approval number: 1408014485). Participants earned $10 an hour for their completion of the self-report measures and the experimental task.

### Sample size requirements

Task data was collected for 74 participants. One possible reaction to the current sample size is to question whether there is sufficient "power" to address the study goals, as smaller sample sizes can lead to greater uncertainty and lower probabilities of detecting an effect. Sample size calculations from other studies that used similar study designs and analytic approaches (i.e., a hierarchical Bayesian regression) show that more than 42 participants can still provide evidence for an effect 64% of the time [40]. From a Bayesian perspective, uncertainty can be quantified by looking at the posterior distribution and assessing the width of the highest density intervals or credible intervals, which can provide graded evidence on

how generalizable the results are [41]. Thus, inferences can still be made from smaller samples, acknowledging that they tend to come with greater degrees of parameter uncertainty that are quantified intuitively with a Bayesian approach.

### Measures

**Socioeconomic measures.** *Annual Household Income:* Household income was measured on an ordinal scale. Participants were asked about their annual household income, with 1 representing $0-$15,000, 2 = $15,001-$30,000, 3 = $30,001-$45,000, 4 = $45,001-$60,000, and 5 = $60,000 +. All sources of income for the household were included in this self-reported measure.

*Area Deprivation Index:* The Area Deprivation Index (ADI) is a spatially derived index that scores census block groups (or "neighborhoods") with a percentile score [42]. This score is a linear combination of 17 U.S. Census-based poverty, education, housing, and employment indicators to rank neighborhoods on their socioeconomic disadvantage [43]. Neighborhoods are ranked from the 1st percentile (having the least socioeconomic disadvantage) to the 100th percentile (having the most socioeconomic disadvantage). The scale of the ADI is in percentiles, with the national average being represented in the 50th percentile. Each participant was given a score on the ADI based on their current address.

*Ambiguity Decision-Making Task:* Participants completed a computerized decision-making paradigm in which the amounts of favorable and unfavorable information about a financial prospect were manipulated [3,44]. On each trial, participants were presented with a virtual "bag" of exactly 100 poker chips, all of which were either red or blue. Participants were asked to indicate their willingness to pay (WTP) for a "red" ticket to play a game in which a single chip is drawn from the bag. If the selected chip was red, the participant won $30, if it was blue, the participant won nothing (and would lose the ticket price). While making this choice, participants received variable information about the number of red and blue chips in the bag (see Fig 1).

Parametrically varying the number of red chips and blue chips shown to participants allowed for a trial-wise calibration of the availability of favorable or unfavorable information. Since red was the "winning" color, red chips represented favorable information in the context of the task, whereas blue chips represented unfavorable information. Grey chips represented ambiguous information, as they were either red (i.e., the "winning" color) or blue (i.e., the "losing" color), but participants did not know which one. The task consisted of 45 trials, in which the number of red chips ranged from 0 (representing a 0% known chance of winning) to 49 (representing a 49% known chance of winning) across trials. The number of grey chips ranged from 5 (representing little ambiguity) to 100 (representing complete ambiguity). Each trial began with a fixation cross, after which participants viewed the available information about the bag's contents. Participants rated their WTP for a red lottery ticket for the current round (range: $0–16) by sliding a marker across a rating bar displayed at the bottom of the screen. Participants were not placed under time constraints for responding.

### Data analysis

Data analysis occurred in two stages: 1) fitting and comparing measurement models of decision-making under uncertainty and 2) testing the association between socioeconomic factors with risk propensity and ambiguity aversion from the best fitting model. The two stages are based on recommendations on model development and measurement validation [19,45], which state that researchers should find the best-fitting measurement model before testing subsequent hypotheses.

We estimate both types of models within a hierarchical Bayesian analysis framework. Hierarchical Bayesian analysis simultaneously estimates group and individual-level parameters using partial pooling, in which the group means constrain and inform each person's ambiguity and risk parameters [46]. Hierarchical models can improve the precision of individual-level estimates by using other sources of information (e.g., the group average or predictors such as socioeconomic factors) to inform estimates of the model parameters [46]. Thus, hierarchical Bayesian analysis provides a flexible framework that allows us to use multiple sources of information to understand decision-making processes.

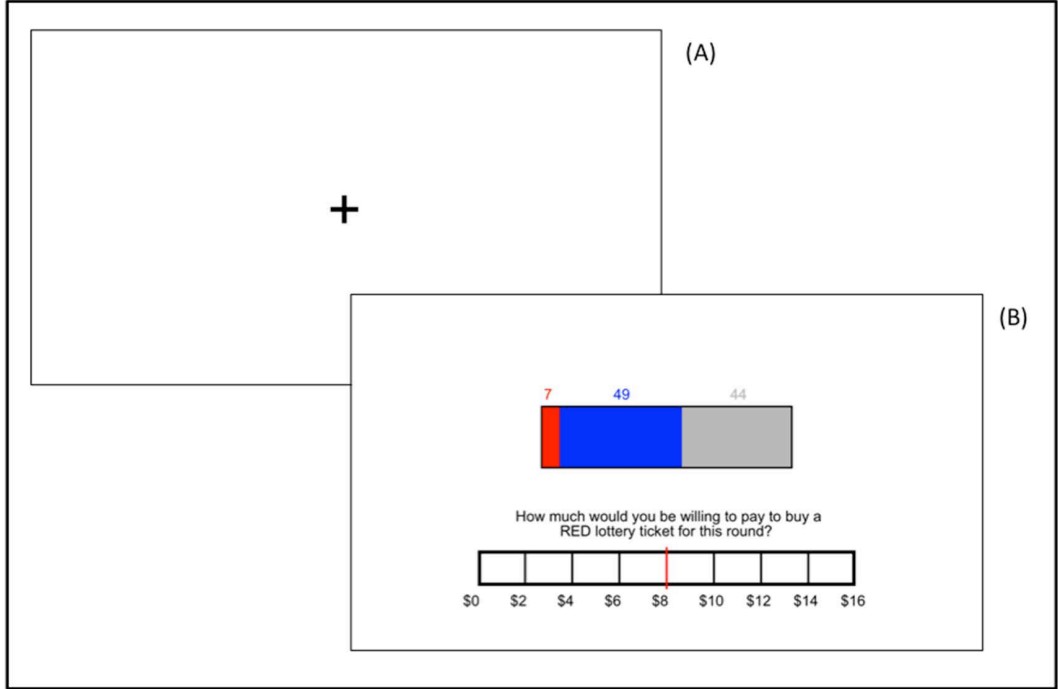

**Fig 1. Ambiguity task diagram.** After a brief fixation period (A), participants were presented with a virtual "bag" of exactly 100 poker chips, all of which were colored either red or blue. On each trial, participants receive information about the number of red (winning) and blue (losing) chips. Participants were asked to indicate their willingness to pay (WTP) for a red ticket (B). In this sample trial, there are 7 red (winning) chips visible, 49 blue (losing) chips visible, and 44 chips that are not visible and thus not known to the participant. Participants selected their WTP by moving a marker across the response bar.

**Fitting and comparing models.** We fit four models that conceptualize decision-making under uncertainty in different ways. We refer to models that are atheoretical and use linear regression as descriptive models. Models derived from existing theory (e.g., expected utility theory) are referred to as theoretical models. We fit each model using the brms R package [47], which uses Stan, a probabilistic programming language for Bayesian inference [48]. We fit both models with a Hamiltonian Markov Chain Monte Carlo estimation procedure [49]. Each model was fit using 6 chains that each contained 3000 warmup samples and 2000 iterations, resulting in a total of 12,000 posterior samples once combining all chains. All chains converged with acceptable model diagnostics (e.g., E-BFMI > 0.2, R-hat < 1.05, ESS > 1000) [19]. We used diffuse priors for the regression weights of each socioeconomic variable ($\beta \sim N(0,10)$) and on both the risk and ambiguity parameters ($\alpha$ andh $\lambda \sim N(0,10)$). These priors reflect having the data influence the results more than any potential prior beliefs we hold about the true effect of each variable. Given the limited research in this area, having strong priors would not be feasible.

Table 1 provides all model equations, and the supporting information (SI) contains additional models, such as those incorporating self-report data on the perceived likelihood of winning for each trial (see SI1 Table in S1 Appendix). To compare models in a Bayesian framework, we use the leave-one-out information criterion (LOOIC), which tells us which model best accounts for the data while penalizing model complexity [50,51]. LOOIC provides estimates of the out-of-sample prediction accuracy and tends to be more robust in finite (i.e., smaller samples) with weak priors or outlier observations [52].

***Baseline descriptive random intercepts model:*** We fit a three-predictor Bayesian hierarchical regression model with random intercepts for each participant. The Bayesian hierarchical regression can be expressed as the following:

$$WTP_{ij} = \beta_{0j} + \beta_1 Red_{ij} + \beta_2 Grey_{ij} + \beta_3 Grey_{ij} \times Red_{ij} + \varepsilon_j \tag{1}$$

**Table 1. All model equations.**

| Model Name | Equation |
|---|---|
| Baseline Descriptive (random intercepts only) Model | $WTP_{ij} = \gamma_{00} + b_{0j} + \beta_1 Red_{ij} + \beta_2 Grey_{ij} + \beta_3 Grey_{ij} \times Red_{ij} + \varepsilon_j$ |
| Maximal Descriptive (random intercepts and slopes) Model | $WTP_{ij} = \gamma_{00} + b_{0j} + (\gamma_{01} + b_{1j} Red_{ij}) + (\gamma_{02} + b_{2j} Grey_{ij}) + (\gamma_{03} + b_{3j} Grey_{ij} \times Red_{ij}) + \varepsilon_j$ |
| Theoretical Linear Subjective Value Model | $WTP_{ij} = (p_{ij} - \lambda_i \frac{A_{ij}}{2})(R_{ij}{}^{\alpha_i})$ |
| Theoretical Exponential Subjective Value Model | $WTP_{ij} = (p_{ij}{}^{(1-\lambda_i A_{ij}}(R_{ij}{}^{\alpha_i})$ |

Where $\varepsilon_j$ is assumed to be independent and identically distributed.

$$\beta_{0j} = \gamma_{00} + b_{0j} \tag{2}$$

Here, WTP for the $i^{th}$ trial within the $j^{th}$ person is a function of the intercept, $\beta_{0j}$, the number of red chips (i.e., the amount of favorable information), the number of grey chips (i.e., the amount of ambiguous information), and the interaction between the red and grey amount (i.e., the interaction between favorable and ambiguous information). We did not include the number of blue chips (i.e., the amount of unfavorable information) in the model because it covaries perfectly with the number of red and grey chips as they all must sum to 100 chips on each trial. The intercept is allowed to vary for each person, suggesting that people differ in their average WTP.

**Maximal descriptive (random intercepts and slopes) model:** We fit a three-predictor Bayesian hierarchical regression model with random intercepts and random slopes for each participant. The random intercepts and slopes regression can be expressed as the following:

$$WTP_{ij} = \beta_{0j} + \beta_{1j} Red_{ij} + \beta_{2j} Grey_{ij} + \beta_{3j} Grey_{ij} \times Red_{ij} + \varepsilon_j \tag{3}$$

Where $\varepsilon_j$ is assumed to be independent and identically distributed.

$$\beta_{0j} = \gamma_{00} + b_{0j} \tag{4}$$

$$\beta_{1j} = \gamma_{01} + b_{1j} \tag{5}$$

$$\beta_{2j} = \gamma_{02} + b_{2j} \tag{6}$$

$$\beta_{3j} = \gamma_{03} + b_{3j} \tag{7}$$

Here, each parameter ($\beta_{1j}$, $\beta_{2j}$, and $\beta_{3j}$) is allowed to vary for each person, suggesting that people differ in their sensitivity to favorable information, ambiguous information, and the interaction between favorable and ambiguous information. We refer to the parameter next to the red chip variable (i.e., $\beta_{1j}$) as the risk parameter and the parameter next to the grey chip variable (i.e., $\beta_{2j}$) as the ambiguity parameter. We deviate from previous literature that uses "risk propensity" and "ambiguity aversion" [53,54] because these parameters come from a descriptive linear regression model and do not have the same theoretical interpretation. However, both parameters can be interpreted as sensitivity to risk and ambiguity. As the risk parameter increases, people are willing to pay more for the lottery when there is more favorable information. As the ambiguity parameter increases, people are willing to pay more when there is more ambiguous information. We present a proof in the SI to show that when the interaction term is 0, the maximal descriptive model is a special case of the theoretical linear subjective value model.

***Theoretical linear subjective value model:*** The theoretical linear subjective value model assumes that each participant's WTP varies as a linear function of the perceived probability of winning and the utility of the reward [11]:

$$WTP = (p - \lambda \frac{A}{2})(R^\alpha)$$

(8)

Where $p$ is the subjective probability of winning the lottery, $\lambda$ is the ambiguity sensitivity parameter, $A$ is the amount of ambiguity, $R$ is the reward amount, and $\alpha$ is the risk propensity parameter. Higher levels of $\lambda$ indicate greater sensitivity to ambiguity, and higher values of $\alpha$ indicate greater risk propensity. WTP is computed for each trial, and the subsequent values are entered into a normal distribution equation to produce the probability of selecting a WTP option:

$$Pr(WTP) = N(\mu_{WTP}, \sigma_{WTP})$$

(9)

$\mu_{WTP}$ represents the average WTP of a person across trials, and $\sigma_{WTP}$ is the variance around their average WTP. Thus, $\sigma_{WTP}$ represents the choice sensitivity and captures how deterministic or how random people's choices are, with low values of $\sigma_{WTP}$ indicating more deterministic choice patterns and higher values indicating more randomness.

***Theoretical exponential subjective value model:*** The theoretical exponential subjective model assumes that each participant's WTP varies as an exponential function of the perceived probability of winning and the utility of the reward [5]:

$$WTP = (p^{1-\lambda A})(R^\alpha)$$

(10)

Where the notation for each variable is the same as the theoretical linear subjective value model and the WTP is assumed to follow a normal distribution.

$\lambda$, $\beta_2$ and $b_{2j}$ (i.e., coefficients next to the grey chip variable) all represent ambiguity aversion. $\alpha$, $\beta_1$ and $b_{1j}$ (i.e., coefficients next to the red chip variable) all represent risk propensity. We use different notations for the descriptive models because they are linear regression models, which typically use $\beta$ and $b$ by convention [55]. For the theoretical models, $\lambda$ and $\alpha$ were used previously to represent ambiguity aversion and risk propensity, respectively [56].

***Best fitting model with socioeconomic predictors:*** To test our hypotheses about the effects of socioeconomic factors on decision-making under uncertainty, we implemented a Bayesian regression by reparameterizing the risk and ambiguity parameters so they are determined by a linear combination of annual household income and ADI [57]. To do so, we first standardized each measure by mean-centering and rescaling by the standard deviation. Next, we estimated deviations in the group-level risk and ambiguity parameters that were attributable to socioeconomic factors using the following regressions:

$$\mu_\lambda = \beta_0 + \beta_1 \times \textit{annual household income} + \beta_2 \times \textit{ADI}$$

(11)

$$\mu_\alpha = \beta_0 + \beta_1 \times \textit{annual household income} + \beta_2 \times \textit{ADI}$$

(12)

Where $\mu_\lambda$ and $\mu_\alpha$ represent the group means for the ambiguity parameter and the risk parameter, respectively. Our modeling of external covariates (i.e., annual household income and ADI) on decision-making processes differs from traditional approaches, which take a single point estimate (e.g., the mean) of a parameter and relate the point estimate to other variables [58]. However, parameters come with uncertainty when they are estimated, which is not accounted for in the traditional approach. Thus, taking point estimates of a parameter can result in overconfident and biased inferences [57]. Incorporating socioeconomic factors as predictors in one hierarchical regression model allows us to directly evaluate how contextual factors are related to parameters (i.e., the risk and ambiguity parameters) in a model while accounting for the uncertainty that is lost when taking point estimates of a parameter. [57].

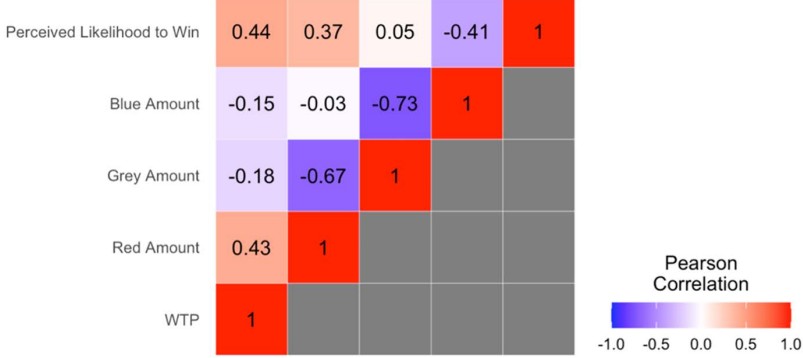

To evaluate the strength of the relationship between the socioeconomic factors and the risk and ambiguity parameters, we used the highest density interval (HDI) + Region of Practical Equivalence (ROPE) procedure [59]. This procedure uses the percentage of the 89% HDI that falls within the ROPE as a decision rule for evaluating whether an effect size is practically far from the null value [60]. If the HDI falls sufficiently outside of the ROPE, then we conclude that there is evidence for an effect. While it is suggested that the HDI and ROPE have zero overlap [59], we interpret the results continuously, such that HDIs with less overlap over the ROPE have more evidence in favor of an effect's existence. Before calculating the ROPE, we z-scored the variables so that the ROPE ranges from −0.1 to 0.1, a small effect size according to Cohen's guidelines [61].

## Results

### Participant socioeconomic characteristics

Forty-two (56.8%) participants reported an annual household income between $0-$15,000, 15 (20.2%) between $15,001-$30,000, 3 (4.1%) between $30,001-$45,000, 4 (5.4%) between $45,001-$60,000, and 10 (13.5%) over $60,000. The mean annual household income was approximately $29,700 (mean = 1.99 [approximately between $15,001-$30,000], SD = 1.44). The ADI score ranged from the 1st to the 98th percentile (mean = 49.23, SD = 23.01).

**Effects of favorable and ambiguous information on WTP.** Across participants, more favorable information (i.e., the number of red chips) was associated with greater WTP ($r = 0.43$) and more unfavorable information (i.e., the number of blue chips) was associated with less WTP ($r = −0.15$). The stronger correlation between favorable information amount and WTP than unfavorable information and WTP suggests that people weigh favorable information more heavily when making decisions. Fig 2 shows that the person's self-reported likelihood of winning is positively correlated with their WTP ($r = 0.44$).

### Model comparison

Table 2 presents each model and the associated LOOIC. The LOOIC of the baseline regression model (LOOIC = 17381.3, SE = 98.7) is the highest, meaning that it performed the worst in predicting out-of-sample data. The model with the second highest LOOIC is the theoretical exponential subjective value model (LOOIC = 18260.7, SE = 117.5), followed by the theoretical linear subjective value model (LOOIC = 17134.8, SE = 107.7). The best model, according to the LOOIC, is the maximal descriptive model in predicting WTP behavior (LOOIC = 16653.7, SE = 116.6). See Table 3 for the parameter estimates of the maximal descriptive model.

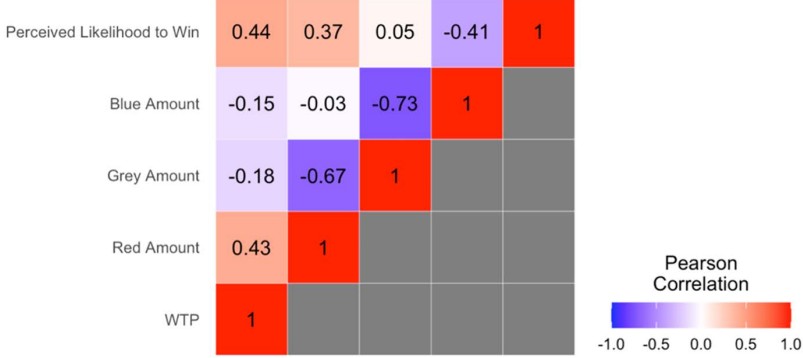

**Fig 2. Correlation matrix of decision-making variables.** WTP = Willingness to Pay.

**Table 2. Model comparison results.**

| Model Name | Leave-One-Out Information Criterion (LOOIC) | Bayesian R² |
|---|---|---|
| Baseline Descriptive (Random Intercepts Only) Model | 17381.3 (98.7) | 0.5 |
| **Maximal Descriptive Model** | **16653.7 (116.6)** | **0.62** |
| Theoretical Linear Subjective Value Model | 17134.8 (107.7) | 0.54 |
| Theoretical Exponential Subjective Value Model | 18260.7 (117.5) | 0.46 |

The bolded model represents the best fitting model based on the leave-one-out information criterion.

**Table 3. Regression results for the maximal descriptive model.**

| Variable | B (SE) | 89% HDI | ROPE % |
|---|---|---|---|
| Intercept | 0.38 (0.51) | −0.43, 1.19 | 54.17% |
| Red Amount | 13.66 (1.15) | 11.81, 15.49 | 0.00% |
| Grey Amount | 2.70 (0.51) | 1.88, 3.51 | 0.00% |
| Red Amount * Grey Amount | 4.09 (2.06) | 0.77, 7.32 | 1.62% |

## Model comparison embedding socioeconomic factors

To test the effect of embedding socioeconomic factors within decision-making models of uncertainty, we regressed annual household income and ADI for each measurement model. Adding the socioeconomic variables improved the LOOIC of the model, LOOIC = 15858.1 (SE = 111.7), meaning that adding annual household income and ADI improves the out-of-sample prediction for WTP. As shown in Table 4, we consistently found that the overall out-of-sample prediction of WTP was improved when adding the socioeconomic factors.

**Risk and ambiguity parameters.** To understand the properties of the risk and ambiguity parameters, we inspected their posterior distributions in the maximal descriptive model by visualizing their distributions and assessing the mean and variance (see Fig 3). The mean value for the risk parameter was positive (mean = 13.66, 89% HDI [11.81, 15.49]), indicating that overall, participants are risk averse. Consistent with previous research, participants are willing to pay more for a lottery when there is more favorable information, and their willingness to pay decreases when there is less favorable information. The mean value of the ambiguity parameter was positive (mean = 2.7, 89% HDI [1.88, 3.51]), indicating that overall, participants are ambiguity-seeking and are willing to pay slightly more for a lottery when there is more unknown information.

## Regressing socioeconomic factors onto risk and ambiguity

We then embedded the two socioeconomic factors (i.e., annual household income and ADI) as predictors of the risk and ambiguity parameters in the maximal descriptive model. Table 5 presents the full results for the socioeconomic maximal descriptive model. For this model, there is evidence that higher annual household income is negatively associated with the ambiguity parameter ($\beta$ = −0.18, 89% HDI [−0.34, −0.02], ROPE = 20.08%) but less evidence that the ambiguity parameter is associated with ADI ($\beta$ = 0.01, 89% HDI [−0.18, 0.19], ROPE = 65.52%). As annual household income increases, the

**Table 4. Model performance for all models with socioeconomic predictors.**

| Model Name | Leave-One-Out Information Criterion (LOOIC) | Bayesian R² |
|---|---|---|
| Socioeconomic Baseline Descriptive (Random Intercepts Only) Model | 16494.0 (95.3) | 0.53 |
| **Socioeconomic Maximal Descriptive Model** | **15858.1 (111.7)** | **0.61** |
| Socioeconomic Theoretical Linear Subjective Value Model | 16285.5 (104.2) | 0.53 |
| Socioeconomic Theoretical Exponential Subjective Value Model | 16635.5 (120.8) | 0.5 |

The bolded model represents the best fitting model based on the leave-one-out information criterion.

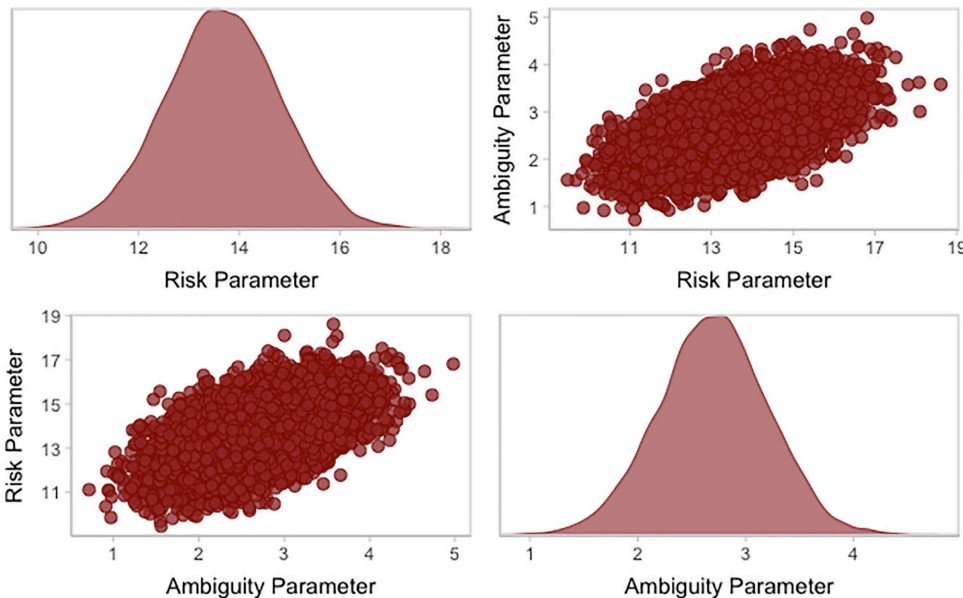

**Fig 3. Posterior distributions of both the risk parameter and ambiguity parameter for the maximal descriptive model, along with a scatterplot of the covariance between the parameters.**

**Table 5. Regression results for the socioeconomic maximal descriptive model.**

| Independent Variable | Dependent variable | β (SE) | 89% HDI | ROPE % |
|---|---|---|---|---|
| Intercept | Risk parameter | 2.90 (0.26) | 2.48, 3.32 | 0% |
| ADI | Risk parameter | −0.18 (0.28) | −0.63, 0.26 | 24.12% |
| Annual Household Income | Risk parameter | 0.12 (0.24) | −0.27, 0.51 | 29.94% |
| Intercept | Ambiguity parameter | 0.60 (0.12) | 0.41, 0.79 | 0% |
| ADI | Ambiguity parameter | 0.01 (0.11) | −0.18, 0.19 | 65.52% |
| **Annual Household Income** | **Ambiguity parameter** | **−0.18 (0.10)** | **−0.34, −0.02** | **19.63%** |
| Intercept | Risk x Ambiguity parameter | 0.81 (0.47) | 0.07, 1.56 | 4.10% |
| ADI | Risk x Ambiguity parameter | −0.65 (0.51) | −1.46, 0.17 | 6.79% |
| Annual Household Income | Risk x Ambiguity parameter | 0.28 (0.45) | −0.43, 1.00 | 15.01% |

The bolded variable represents evidence of an effect based on the 89% HDI.

degree of ambiguity aversion increases, suggesting that those with lower income are more ambiguity seeking, and those with higher income tend to be more ambiguity averse.

The relationship between all socioeconomic variables and the risk parameter was weak (ADI: $\beta$ = −0.18, 89% HDI [−0.63, 0.26]; Annual Household Income: $\beta$ = 0.12, 89% HDI [−0.27, 0.51], as all HDIs overlapped with 0, suggesting that there is little evidence of an effect between the socioeconomic variables and risk sensitivity. The relationship between the socioeconomic variables and the interaction of favorable and ambiguous information was also weak, as the HDIs and ROPE interval suggested little evidence for an effect.

Given that all variables were standardized, we see that some HDIs vary widely (e.g., ADI and the Risk x Ambiguity parameter ranges from −1.46 to 0.17), suggesting that the relationship could be strongly negative or slightly positive. Other HDIs (e.g., annual household income and the Ambiguity parameter, 89% HDI [−0.34, −0.02], provide more certainty about the direction and strength of the effect, showing that it is likely a modest negative relationship. See Fig 4 for

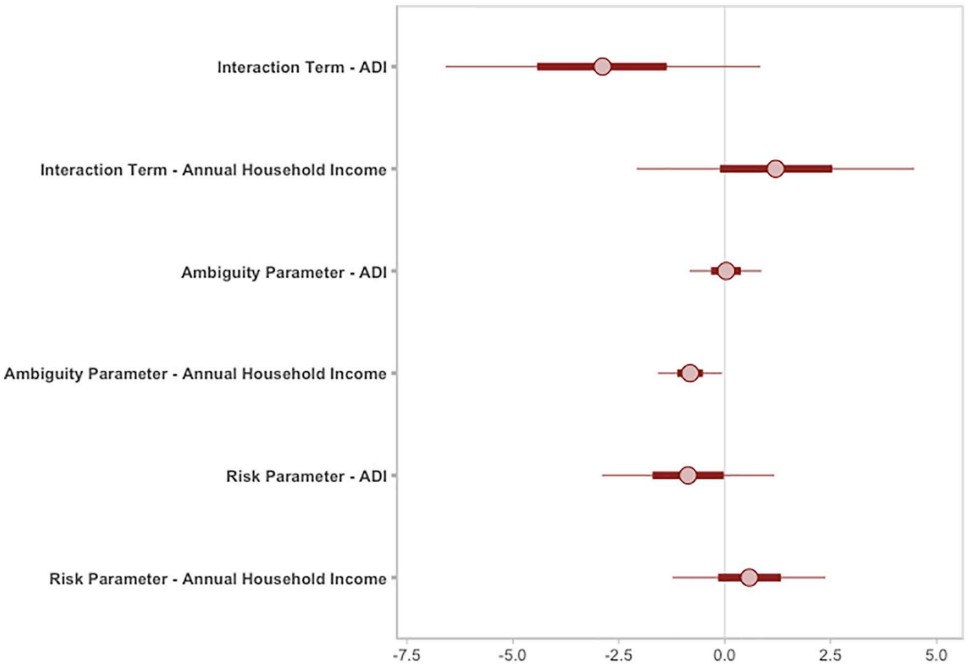

**Fig 4. 89% Highest Density Intervals (HDI) for the regression weights of each socioeconomic variable on the risk parameter and ambiguity parameter.** ADI = Area Deprivation Index.

the HDIs of each fixed-effects regression coefficient. To better understand the independent effect of each socioeconomic predictor we analyzed regression models in which annual household income and ADI were entered as the sole predictor of the risk and ambiguity parameters in separate models. We present the results for these regression models in SI2 Table and SI3 Table in S1 Appendix.

## Discussion

Decision-making under uncertainty is a common phenomenon, making it essential to develop models that accurately estimate and analyze the underlying processes, such as risk and ambiguity. Descriptive models use linear regression to explain people's decisions, while theoretical models leverage economic theory (e.g., expected utility theory) to model decision-making. To date, no studies have directly compared descriptive and theoretical models of decision-making under uncertainty. Failing to compare measurement models assumes that researchers know the most appropriate model of cognitive processes, an assumption that can run the risk of not accurately representing cognition within certain samples. We consider several models that represent decision-making in different ways, such as linear regression models that account for the interaction between risk and ambiguity. We find that the maximal descriptive model provides the best out-of-sample prediction for participants' WTP. The maximal descriptive model is a random intercept and slopes linear regression, allowing each person to have a risk and ambiguity parameter that deviates from the sample mean. Further, we find that adding socioeconomic predictors of the risk and ambiguity parameters improves the out-of-sample prediction ability. Finally, we find that annual household income is associated with the ambiguity parameter, such that those with lower annual household income tend to be more ambiguity-seeking. Together, these findings demonstrate that a descriptive model provides insight into how people's WTP varies as a linear combination of risk, ambiguity, and the interaction of their risk and ambiguity. Moreover, these findings show that contextualizing measurement models within their socioeconomic background can improve models of decision-making.

The model-building and evaluation process in the present study demonstrated that linear regression can adequately explain WTP within a decision-making task under uncertainty. Linear regression serves as a useful starting point for understanding which variables explain people's decisions and provides an interpretable way to understand the data [19]. To this end, we fit hierarchical linear regression models, starting with a random intercepts-only model, followed by a random intercepts and slopes model. The best-fitting model (i.e., the maximal descriptive model) suggests that WTP slightly increased as the amount of ambiguity increased. Moreover, consistent with previous work [62], WTP tended to increase when the amount of risk decreased. The maximal descriptive model differs from the theoretical models in that it captures the influence of the interaction between favorable and ambiguous information on participants' WTP, which may indicate that the presence of both favorable and ambiguous information in the same trial affects people's decisions. Importantly, these findings do not mean that other models of decision-making under uncertainty are incorrect. They indicate that it is worth testing assumptions about which model is most appropriate for a given sample. However, a balance needs to be struck between a "one-size-fits-all" approach in which one model is applied to an entire population and having different measurement models for each sample. More studies should test different models and observe if there is a model that consistently emerges as a better measurement model across studies. The present study is a proof-of-concept of the importance of comparing measurement models. We highlight that a previous unobserved representation of decision-making under uncertainty best characterized processes in the present sample.

From the modeling perspective, there are some important limitations in the present study to note. Here, we use a smaller sample size compared to other studies that conduct model comparisons [63–65], which could limit inference and the generalizability of the results. However, the present sample is diverse in terms of its socioeconomic background and lived experiences. Attaining more diverse samples is important for making inferences about the population, yet it is often more difficult when targeting people of different socioeconomic backgrounds [66,67]. Moreover, the HDI estimates did not vary widely (e.g., from very large positive effect sizes to very large negative effect sizes), suggesting that our estimates provide some amount of certainty. Finally, the Bayesian framework allows other researchers to use the posterior distribution from the current study as a starting point for priors in future work. The current study can directly inform future analyses of socioeconomic factors and decision-making under uncertainty.

In addition to establishing robust model development workflows, model development procedures should consistently consider contextual factors that influence underlying decision-making processes. Decision-making does not occur in a vacuum. In the present study, we find that adding socioeconomic factors (i.e., annual household income and neighborhood disadvantage) improves the model. The leave-one-out information criterion for the maximal descriptive model improves with the addition of socioeconomic factors, suggesting that contextualizing decision-making enhances the prediction of people's decisions in the financial decision-making task under uncertainty. We find that higher ambiguity aversion is associated with higher annual household income, and those with lower annual household income are less averse to ambiguity. Unlike previous research [68,69], we did not see evidence of a relationship between socioeconomic factors and risk sensitivity. It could be that once accounting for ambiguity sensitivity, socioeconomic factors do not affect people's risk sensitivity, suggesting that different types of uncertainty should be accounted for when studying decision-making.

Consistent with previous research [29], we find that higher ambiguity aversion is associated with higher household income, whereas those with lower household income were less averse to ambiguity. Those with fewer resources in the household may be habituated to an environment of ambiguity in their everyday lives, which is reflected in their reduced sensitivity to ambiguity within the experimental task [70,71]. Among individuals with limited household resources, chronic exposure to adversity may attenuate the perceived significance of minor financial losses. This diminished sensitivity may stem from the prioritization of handling more immediate life stressors or from an internalized expectation that financial setbacks are a normative aspect of daily life [70–72]. Moreover, those with lower annual household income may be more motivated to pursue rewards despite the uncertainty, given the tenuous nature of their financial circumstances [73].

Of note, the annual household income-ambiguity effect reported here is counter to the research on the decreasing absolute ambiguity aversion assumption [74–76], which posits that individuals become less ambiguity averse as their wealth increases [30,31,74,77]. However, much of the previous work supporting decreasing absolute ambiguity aversion was conducted in samples with little representation of lower income participants, unlike the present study and other studies [28,29] conducted in developing countries. Further, there is growing evidence that decreasing absolute ambiguity aversion is far from a settled conclusion, with the specifics of one's socioeconomic background or other individual characteristics playing a key role in contextualizing decision-making effects [78–81]. Future research can continue to test the bounds of the decreasing absolute ambiguity aversion assumption, seeking to understand in what contexts and for whom this assumption is most relevant. Future research also should investigate the mechanisms for people's blunted ambiguity when they have a lower household income. Notably, there was a stronger relationship between annual household income and ambiguity than with ADI and ambiguity. It could be that proximal factors (e.g., struggling to pay rent) that are more closely tied to lower household income affect one's decision-making compared to surrounding neighborhood-level factors, which are more distal.

The current study found that annual household income may influence people's consideration of ambiguity. However, it is important to note the limitations of our approach. First, we only assessed current socioeconomic status. However, a person's socioeconomic background can change over time, with earlier experiences and socioeconomic upbringing playing an important role in one's growth and development [82,83]. Future research should assess the influence of one's full socioeconomic history on one's decision-making or assess the effects of one's current socioeconomic status over and above one's childhood socioeconomic background. Second, the relationship between annual household income and the ambiguity parameter is considered modest by conventional guidelines in psychology ($\beta = 0.17$) [61]. To evaluate the practical meaningfulness of the finding, future research should assess whether the ambiguity parameter in the socioeconomic maximal descriptive model corresponds to real-world ambiguous behaviors (e.g., theft, financial investments). Finally, while the current study assessed a person's socioeconomic conditions, it did not directly assess the harshness or unpredictability of their environment. According to life history theory, environmental unpredictability and harshness can lead people to prioritize larger and less certain rewards when they live on a "faster course" and have less time to acquire smaller but safer rewards [84]. Future research can directly measure environmental harshness and unpredictability and assess whether these are mechanisms underlying people's decision-making tendencies under ambiguity.

The model validation process is important because it checks our assumptions that one model is the most appropriate for describing decision-making. In this study, a hierarchical linear regression model is sufficient to represent people's willingness to pay when deciding under uncertainty. Moreover, including socioeconomic factors in the model can help improve prediction. Finally, lower household income, in particular, is associated with less ambiguity aversion. These findings demonstrate the value of methodically choosing the best model of decision-making and the importance of embedding a person's decisions within the appropriate socioeconomic context.

## Supporting information

**S1 Appendix. Testing additional measurement models and reparameterizing the linear subjective value model.** This document includes methods and results for the self-report informed measurement models of decision-making under uncertainty. Moreover, we provide regression results with ADI and annual household income as the sole predictors of risk and ambiguity. Lastly, we present a proof that reparametrizes the linear subjective value model as a random intercepts and slopes linear regression.
(DOCX)

## Author contributions

**Conceptualization:** Brendan Lam, Hunter Robbins, Arielle Baskin-Sommers.

**Data curation:** Brendan Lam, Arielle Baskin-Sommers.

**Formal analysis:** Brendan Lam, Samuel Paskewitz.

**Funding acquisition:** Arielle Baskin-Sommers.

**Investigation:** Brendan Lam, Arielle Baskin-Sommers.

**Methodology:** Brendan Lam, Samuel Paskewitz, Arielle Baskin-Sommers.

**Project administration:** Brendan Lam, Arielle Baskin-Sommers.

**Resources:** Brendan Lam, Arielle Baskin-Sommers.

**Software:** Brendan Lam, Samuel Paskewitz.

**Supervision:** Samuel Paskewitz, Arielle Baskin-Sommers.

**Validation:** Brendan Lam, Arielle Baskin-Sommers.

**Visualization:** Brendan Lam.

**Writing – original draft:** Brendan Lam, Hunter Robbins, Arielle Baskin-Sommers.

**Writing – review & editing:** Brendan Lam, Samuel Paskewitz, Arielle Baskin-Sommers.

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
