## [Decision Letter · Decision Letter 0]

11 Jul 2025

Dear Dr. Lam,

Thank you for submitting your manuscript to PLOS ONE. After careful consideration, we feel that it has merit but does not fully meet PLOS ONE’s publication criteria as it currently stands. Therefore, we invite you to submit a revised version of the manuscript that addresses the points raised during the review process.

We look forward to receiving your revised manuscript.

Kind regards,

Pablo Gutierrez Cubillos

Academic Editor

PLOS ONE

Journal Requirements:

“This research was supported in part by grants through the American Psychological Foundation, American Psychology-Law Society, and Harry F. Guggenheim Foundation (PI: Baskin-Sommers).”

4. In the online submission form you indicate that your data is not available for proprietary reasons and have provided a contact point for accessing this data. Please note that your current contact point is a co-author on this manuscript. According to our Data Policy, the contact point must not be an author on the manuscript and must be an institutional contact, ideally not an individual. Please revise your data statement to a non-author institutional point of contact, such as a data access or ethics committee, and send this to us via return email. Please also include contact information for the third party organization, and please include the full citation of where the data can be found.

6. We notice that your supplementary tables are included in the manuscript file. Please remove them and upload them with the file type 'Supporting Information'. Please ensure that each Supporting Information file has a legend listed in the manuscript after the references list.

Additional Editor Comments (if provided):

Dear Mr. Lam,

Thank you for submitting your interesting research to PLOS ONE. As you can see, both referees have engaged constructively with your work, and after my own review, I concur with their assessments.

That said, I encourage you to strengthen the connection between your empirical findings and the economic modeling of ambiguity and risk. In particular, standard economic theory often relies on the assumptions of decreasing absolute risk aversion and decreasing absolute ambiguity aversion. Your results, however, suggest that poorer individuals invest a larger share of their income in ambiguous assets or activities than wealthier individuals. This pattern appears to contradict widely accepted theoretical predictions. In this context, I would appreciate a more explicit discussion of how your findings challenge the conventional framework and what implications this has for the behavioral assumptions underpinning economic models of decision-making under uncertainty.

Please respond carefully to all referee comments as well as to the point raised above.

Best regards,

Reviewers' comments:

Reviewer's Responses to Questions

**Comments to the Author**

1. Is the manuscript technically sound, and do the data support the conclusions?

Reviewer #1: Yes

Reviewer #2: Yes

2. Has the statistical analysis been performed appropriately and rigorously?

Reviewer #1: Yes

Reviewer #2: Yes

3. Have the authors made all data underlying the findings in their manuscript fully available?

Reviewer #1: No

Reviewer #2: Yes

4. Is the manuscript presented in an intelligible fashion and written in standard English?

Reviewer #1: Yes

Reviewer #2: Yes

Reviewer #1: Dear Author, The manuscript makes a valuable contribution to the field and presents its findings clearly and effectively.

In future author should concentrate on more innovative work rather than comparison of existing ones.

Reviewer #2: The paper entitled “Comparing Descriptive and Theoretical Models of Decision-Making Under Uncertainty and Their Relation to Socioeconomic Factors” has a relevant research topic. The researchers performed a comparative analysis of four models for assessing decision-making under uncertainty, which differ in the mathematical form of risk and ambiguity, to determine which one best predicts willingness-to-pay behavior in a financial decision-making task.

The methods were well detailed. However, I make some considerations in order to improve the reader's understanding.

(1) Regarding the scale of the socioeconomic measure “Household Income”, is the Household Income considered monthly or annual? I suggest specifying this in the text.

(2) The abstract is well structured and the research well defined. However, the authors do not mention the methodology used to perform the comparative analysis. They also do not mention the limitations of the research.

(3) Although the introduction reviews the types of models and describes the objective of the research, the methodology used is not mentioned (which methodology was used to build the models and to compare and evaluate the models constructed).

(4) The figures and tables are in agreement. However, on pages 17 and 18, the text cites a value for LOOIC of the Theoretical Exponential Subjective Value Model and another value is presented in Table 2. Both Table 2 and Table 4 shows a higher LOOIC for the Theoretical Exponential Subjective Value Model. I suggest checking this result.

(5) Page 10 has a misconfigured sentence in the last paragraph.

(6) On page 14, in “Theoretical linear subjective value model” the authors state that “Higher levels of “beta” indicate greater sensitivity to ambiguity, (...)”, however, “lambda” is the ambiguity sensitivity parameter. This needs to be checked.

(7) On page 12, in “Baseline descriptive random intercepts model”, 0 is given in Equation (2). In Table 1, this parameter was represented differently than what was agreed in Equation (2). I suggest keeping the same representation to avoid ambiguous information.

(8) On page 16, Equations (11) and (12) are the same. Is this correct?

**Do you want your identity to be public for this peer review?** For information about this choice, including consent withdrawal, please see our Privacy Policy

Reviewer #1: **Yes: ** Dr. Nancy

Reviewer #2: No

---

## [Author Response · Author response to Decision Letter 1]

15 Aug 2025

Editor Comments

Authors’ response: We reviewed the style requirements and confirm that our submission conforms to the template.

“This research was supported in part by grants through the American Psychological Foundation, American Psychology-Law Society, and Harry F. Guggenheim Foundation (PI: Baskin-Sommers).”

Authors’ response: Thank you for letting us know. We included an amended funding statement in a revised cover letter and uploaded it to the resubmission portal.

Cover letter: This research was supported in part by grants through the American Psychological Foundation, American Psychology-Law Society, and Harry F. Guggenheim Foundation (PI: Baskin-Sommers). There was no additional external funding received for this study.

Authors’ response: We received permission to upload a deidentified dataset. We updated our Data Availability statement.

4. In the online submission form you indicate that your data is not available for proprietary reasons and have provided a contact point for accessing this data. Please note that your current contact point is a co-author on this manuscript. According to our Data Policy, the contact point must not be an author on the manuscript and must be an institutional contact, ideally not an individual. Please revise your data statement to a non-author institutional point of contact, such as a data access or ethics committee, and send this to us via return email. Please also include contact information for the third party organization, and please include the full citation of where the data can be found.

Authors’ response: We revised our data statement to a non-author institutional point of contact: Yale University Institutional Review Board (HRPP@yale.edu).

Authors’ response: We revised our ethics statement in the Methods section, which specifies the full name of the IRB committee that approved the study.

Page 9: All participants provided written informed consent. All procedures were completed in line with the protocol approved by the Yale University Institutional Review Board (approval number: 1408014485).

6. We notice that your supplementary tables are included in the manuscript file. Please remove them and upload them with the file type 'Supporting Information'. Please ensure that each Supporting Information file has a legend listed in the manuscript after the references list.

Authors’ response: We deleted the supplementary tables from the manuscript file. They are now solely in the ‘Supporting Information’ file. The Supporting Information file has a legend in the manuscript after the references list (page 34).

Authors’ response: There were no specific citation recommendations.

Authors’ response: We reviewed our reference list for accuracy.

9. That said, I encourage you to strengthen the connection between your empirical findings and the economic modeling of ambiguity and risk. In particular, standard economic theory often relies on the assumptions of decreasing absolute risk aversion and decreasing absolute ambiguity aversion. Your results, however, suggest that poorer individuals invest a larger share of their income in ambiguous assets or activities than wealthier individuals. This pattern appears to contradict widely accepted theoretical predictions. In this context, I would appreciate a more explicit discussion of how your findings challenge the conventional framework and what implications this has for the behavioral assumptions underpinning economic models of decision-making under uncertainty.

Authors’ response: Thank you for the suggestion. We edited the introduction and the discussion to address the point. In the introduction, we detail the mixed literature on the association between ambiguity and decision-making (page 6). In the discussion, we added a section on the implications of our findings for standard economic theory and decreasing absolute ambiguity aversion.

Page 6: Research on the relationship between socioeconomic factors and ambiguity aversion has been mixed (28,29). For example, studies using hypothetical monetary gambles or hypothetical financial investments in samples of Dutch participants (with average annual household incomes of €35,256 and €45,792, respectively) (30,31) show that individuals with higher economic resources display lower ambiguity aversion (30,31) (i.e., decreasing absolute ambiguity aversion), that is, people with more wealth appear more willing to make decisions with uncertain outcomes (e.g., investing in education or the stock market) for potentially higher returns (32). However, research conducted in developing countries using ambiguous phrases, field experiments, or a two-choice decision task tends to find different patterns for the relationship between ambiguity and decision-making depending on the experimental context. One study using ambiguous phrases shows that people living in urban areas with lower economic resources were more ambiguity seeking, whereas those living in rural areas with lower economic resources were more averse to ambiguity (28,29). Importantly, though, the effects were modified by the relative level of household income, such that within lower economic resourced families, those that had more resources show more sensitivity to ambiguity. This suggests that for the poorest people, the certainty of uncertainty (33) may lead them to be less adverse to ambiguous probabilities (34). Other studies using field experiments (e.g., distribution of crops) or a two-choice decision task show that relative income level among those in developing countries did not relate to ambiguity sensitivity (28). To date, though, no studies first compared different measurement models of decision-making before testing the relationship between socioeconomic resources and decision-making under uncertainty. This means that the mixed findings within this literature could, in part, reflect a failure to accurately estimate the decision processes of risk and ambiguity when put into context of real-world resources.

Pages 25-26: Consistent with previous research (29), we find that higher ambiguity aversion is associated with higher household income, whereas those with lower household income were less averse to ambiguity. Those with fewer resources in the household may be habituated to an environment of ambiguity in their everyday lives, which is reflected in their reduced sensitivity to ambiguity within the experimental task (70,71). Among individuals with limited household resources, chronic exposure to adversity may attenuate the perceived significance of minor financial losses. This diminished sensitivity may stem from the prioritization of handling more immediate life stressors or from an internalized expectation that financial setbacks are a normative aspect of daily life (70–72). Moreover, those with lower annual household income may be more motivated to pursue rewards despite the uncertainty, given the tenuous nature of their financial circumstances (73).

Of note, the annual household income-ambiguity effect reported here is counter to the research on the decreasing absolute ambiguity aversion assumption (74–76), which posits that individuals become less ambiguity averse as their wealth increases (30,31,74,77). However, much of the previous work supporting decreasing absolute ambiguity aversion was conducted in samples with little representation of lower income participants, unlike the present study and other psychological studies (28,29) conducted in developing countries. Further, there is growing evidence that decreasing absolute ambiguity aversion is far from a settled conclusion, with the specifics of one’s socioeconomic background or other individual characteristics playing a key role in contextualizing decision-making effects (78–81). Future research can continue to test the bounds of the decreasing absolute ambiguity aversion assumption, seeking to understand in what contexts and for whom this assumption is most relevant. Future research also should investigate the mechanisms for people’s blunted ambiguity when they have a lower household income. Notably, there was a stronger relationship between annual household income and ambiguity than with ADI and ambiguity. It could be that proximal factors (e.g., struggling to pay rent) that are more closely tied to lower household income affect one’s decision-making compared to surrounding neighborhood-level factors, which are more distal.

Comments to the Author

1. Is the manuscript technically sound, and do the data support the conclusions?

Reviewer #1: Yes

Reviewer #2: Yes

Authors’ response: We thank the reviewers for noting that our work was technically sound.

2. Has the statistical analysis been performed appropriately and rigorously?

Reviewer #1: Yes

Reviewer #2: Yes

Authors’ response: We thank the reviewers for noting that our analyses were appropriate.

3. Have the authors made all data underlying the findings in their manuscript fully available?

Reviewer #1: No

Reviewer #2: Yes

Authors’ response: We now received approval to upload a deidentified dataset to OSF. All data and code are now publicly available.

4. Is the manuscript presented in an intelligible fashion and written in standard English?

Reviewer #1: Yes

Reviewer #2: Yes

Authors’ response: We thank the reviewers for noting that our work was presented in an intelligible fashion.

5. Review Comments to the Author

Reviewer #1: Dear Author, The manuscript makes a valuable contribution to the field and presents its findings clearly and effectively. In future author should concentrate on more innovative work rather than comparison of existing ones.

Authors’ response: We thank the reviewer for noting that our manuscript makes a “valuable contribution.” We appreciate the encouragement of the reviewer to take the foundation laid by the present study and expand to more innovative models in future work. We see the present study as a necessary step in establishing a solid framework for measurement models of risk and ambiguity in this area that will serve to guide future work.

Reviewer #2: The paper entitled “Comparing Descriptive and Theoretical Models of Decision-Making Under Uncertainty and Their Relation to Socioeconomic Factors” has a relevant research topic. The researchers performed a comparative analysis of four models for assessing decision-making under uncertainty, which differ in the mathematical form of risk and ambiguity, to determine which one best predicts willingness-to-pay behavior in a financial decision-making task. The methods were well detailed. However, I make some considerations in order to improve the reader's understanding.

Authors’ response: We thank the reviewer for noting that our methods were “well detailed” and we appreciate the suggestions for refining our presentation.

---

## [Editor Report · Decision Letter 1]

28 Aug 2025

Comparing Descriptive and Theoretical Models of Decision-Making Under Uncertainty and Their Relation to Socioeconomic Factors

PONE-D-25-16859R1

Dear Dr. Lam,

We’re pleased to inform you that your manuscript has been judged scientifically suitable for publication and will be formally accepted for publication once it meets all outstanding technical requirements.

Kind regards,

Pablo Gutierrez Cubillos

Academic Editor

PLOS ONE

Additional Editor Comments (optional):

Dear Brendan,

We are delighted to accept your paper for publication. We only ask that you replace the phrase “certainty of uncertainty” with “cotidianity of uncertainty”, as the former does not make any theoretical sense.

Best regards,

Pablo
---

## [Editor Report · Acceptance letter]

PONE-D-25-16859R1

PLOS ONE

Dear Dr. Lam,

I'm pleased to inform you that your manuscript has been deemed suitable for publication in PLOS ONE. Congratulations! Your manuscript is now being handed over to our production team.

Kind regards,

on behalf of

Dr. Pablo Gutierrez Cubillos

Academic Editor

PLOS ONE